# Graph-Enhanced EEG-to-Text Decoding: A Spatio-Temporal Relational Embedding Framework for Brain Signal Translation

## Abstract

Despite recent progress in brain–computer interfaces (BCIs), decoding natural language directly from EEG remains a critical challenge. Existing EEG-to-text models primarily treat signals as sequential time series, which severely limits their ability to capture the spatial and temporal relationships among electrodes and limits the possibility of generalization in low-data regimes. To address this challenge, we propose a novel graph-enhanced framework to explicitly model relational information in brain signals. The key idea of our framework is to construct Spectro-Topographic Relational Graphs (STRG) that jointly encode static electrode topology and dynamic inter-channel functional connectivity. From these graphs, we derive Spatio-Temporal Relational Embeddings (STRE), which provide graph-aware representations for downstream sequence-to-sequence decoding. Specifically, (i) STRG captures spatial adjacency and frequency-specific connectivity, (ii) STRE transforms these relational structures into embeddings aligned with text decoding, and (iii) the overall framework integrates these embeddings with a neural decoder to generate natural language outputs. To the best of our knowledge, this is the first graph-enhanced approach for EEG-to-text decoding that explicitly uses graph-based representations of EEG signals. Empirical results show that our framework delivers substantial improvements over strong recurrent and Transformer baselines. In particular, our Graph-Enhanced EEG-to-Text Decoding achieves up to 16% relative gains on BLEU-4, which highlights the effectiveness of relational graph modeling for advancing neural decoding.

## 1 Introduction

### 1.1 Background and Significance of the Study

Decoding natural language directly from brain signals has long been a central goal in brain–computer interfaces (BCIs), with transformative applications for assistive communication and the development of cognitive human–machine interaction (Murad & Rahimi, 2024). Success in this area would provide a direct communication channel for individuals with severe speech or motor impairments, while also advancing fundamental understanding of how language is represented in neural activity (Lamprou & Moshfeghi, 2024). Electroencephalography (EEG), as a non-invasive, cost-effective, and temporally precise technique for recording brain signals, offers a particularly promising avenue toward this goal.

Over the past decade, researchers have begun to adapt models originally designed for natural language processing and speech recognition to the problem of EEG-to-text decoding (Wang et al., 2024). Most existing work relies on sequential encoders such as recurrent neural networks (RNNs) and, more recently, Transformer-based architectures. These models are effective at capturing local and long-range temporal dependencies in EEG signals, which makes them attractive candidates for neural decoding. Nevertheless, compared with other recording modalities such as electrocorticography (ECoG) or fMRI, EEG-to-text decoding remains considerably more challenging. Performance remains limited, which highlights the need for new representational paradigms that are better aligned with the unique characteristics of EEG.

## 1.2 CHALLENGES IDENTIFIED

A central obstacle lies in how existing models conceptualize a central obstacle lies in how existing models conceptualize EEG data. Predominantly, EEG is treated as a sequential time series, processed along the temporal dimension while disregarding the explicit spatial and spatio-temporal relationships among channels (Burle et al., 2015). Although RNNs and Transformers are powerful at capturing temporal patterns, they neglect the topographic organization of the electrodes and the functional connectivity between cortical regions, both of which play a critical role in shaping the neural processes underlying language comprehension.

This limitation manifests in several ways. Models that flatten EEG into purely sequential inputs tend to lose structural information about spatial adjacency and frequency-specific interactions (Yang & Jia, 2024). Such models often fail to generalize in low-data regimes, which is particularly problematic given that EEG-to-text datasets remain relatively small in size (Fastenrath et al., 2009). In addition, because they do not account for inter-channel connectivity, sequential approaches offer little interpretability with respect to how different cortical regions interact during language processing. These challenges collectively hinder progress in building robust and scientifically meaningful EEG-to-text systems.

## 1.3 IMPROVEMENTS AND CONTRIBUTIONS

In this work, we introduce graph-enhanced EEG representation learning as a new paradigm for brain-to-text decoding. Rather than modeling EEG as a simple sequence of time steps, our approach explicitly encodes relational information across electrodes and frequency bands. At the core of our framework are Spectro-Topographic Relational Graphs (STRG), which unify static electrode topology with dynamic functional edges derived from inter-channel correlations. This formulation preserves both the anatomical structure of EEG placement and the functional connectivity that emerges during language processing. From these graphs, we derive Spatio-Temporal Relational Embeddings (STRE), which capture graph-aware representations that integrate spatial adjacency, temporal dynamics, and spectral features in a unified embedding space tailored for sequence-to-sequence text generation (Miri et al., 2024).

By incorporating this relational perspective, our framework provides richer and more interpretable neural representations than conventional sequential models. It also enables stronger generalization in low-data regimes, since relational information offers an inductive bias that constrains the learning process in a principled way. We formalize STRG as a principled graph construction method that unifies electrode topology with dynamic connectivity, and we introduce STRE as a novel embedding mechanism for EEG representation learning. We then integrate these embeddings into a neural decoder to generate natural language outputs.

Empirical evaluation on benchmark datasets such as ZuCo demonstrates that our approach consistently outperforms strong recurrent and Transformer baselines. In particular, the framework achieves up to a 16% improvement on BLEU-4, underscoring the effectiveness of graph-based relational modeling for natural language decoding from EEG. Beyond immediate performance gains, this work establishes a foundation for more robust and interpretable neural decoding systems, opening new avenues for future research at the intersection of brain–computer interfaces, graph representation learning, and natural language processing.

## 2 RELATED WORK

## 2.1 EEG-TO-TEXT DECODING

Early EEG decoding methods relied on statistical learning and shallow machine learning techniques, typically designed for simple classification tasks such as motor imagery (MI), P300 spellers, or emotion recognition. These applications established the feasibility of mapping EEG signals to discrete labels, but offered limited scalability toward natural language generation (Saeidi et al., 2021; Lopez-Bernal et al., 2022).

## 2.2 SEQUENCE-TO-SEQUENCE MODELS

More recent approaches use sequence-to-sequence models such as RNN and Transformer-based decoders, which are well established in natural language processing and have been adapted for EEG-to-text tasks. These models excel at capturing temporal dependencies but inherently assume a linear sequence structure. As a result, they fail to exploit the inter-electrode correlations and spatial dependencies intrinsic to neural activity. In practice, when applied to EEG decoding, these models are prone to overfitting and reduced robustness, particularly in low-data scenarios that are common in neuroscience applications (Murad & Rahimi, 2024; Hollenstein et al., 2021).

## 2.3 GRAPH REPRESENTATIONS IN EEG ANALYSIS

Graph-based methods have become increasingly popular in neuroscience to capture both spatial and functional structure in brain activity. Functional connectivity graphs are often constructed using correlation-based metrics such as Pearson correlation, coherence, or phase-locking value (PLV) (Šverko et al., 2022). Spatial adjacency graphs encode electrode topology, reflecting the physical arrangement of EEG channels. These graph representations have been applied successfully in downstream tasks such as seizure detection, emotion recognition, and brain disorder analysis (Tian & Zhang, 2025; Díaz-Montiel et al.; Abadal et al., 2025). However, prior work has been almost exclusively limited to classification settings. To the best of our knowledge, no existing studies integrate graph structures directly into EEG-to-text decoding. This represents a gap, as most applications of graph modeling in EEG remain disconnected from sequence generation tasks.

# 3 PRELIMINARIES

## 3.1 EEG GRAPH CONSTRUCTION BACKGROUND

Electroencephalography (EEG) records multichannel neural activity from electrodes placed on the scalp. Formally, the EEG input can be represented as:

$$X^{(f)} \in \mathbb{R}^{C \times T}, \quad f \in \{\delta, \theta, \alpha, \beta, \gamma\},$$

where $C$ is the number of channels (electrodes) and $T$ is the number of time steps. Each choice of $f$ corresponds to a canonical EEG frequency band: $\delta$ (Delta, 0.5–4 Hz), $\theta$ (Theta, 4–8 Hz), $\alpha$ (Alpha, 8–12 Hz), $\beta$ (Beta, 13–30 Hz), and $\gamma$ (Gamma, >30 Hz). Each row $x_i \in \mathbb{R}^T$ denotes the time series recorded from electrode $i$. In this way, $X^{(f)}$ provides an electrode-by-time representation restricted to a specific frequency range, preserving both spatial ($C$) and temporal ($T$) structure within that band (Bajaj et al., 2022).

A natural way to represent EEG signals is through graphs that encode both spatial and functional relationships among electrodes. One approach is to construct static topology graphs, where edges reflect electrode adjacency based on physical scalp placement, such as the standard 10–20 system. Complementary to this, functional connectivity graphs capture dynamic interactions by estimating statistical dependencies between electrode signals, commonly using measures such as Pearson correlation. Together, these representations provide complementary views of the data, combining spatial adjacency, dynamic coupling, and frequency-specific features into a unified framework for graph-based modeling (Chao et al., 2023).

Formally, a graph is defined as $G = (V, E)$ where $V$ denotes the set of nodes and $E$ the set of edges. The adjacency matrix $A \in \mathbb{R}^{|V| \times |V|}$ encodes the edge structure, with $A(i, j)$ representing the presence or weight of an edge between nodes $i$ and $j$. The static spatial adjacency can be written as

$$A_{\text{spatial}}(i, j) = \begin{cases} 1, & \text{if electrodes } i, j \text{ are neighbors on the scalp,} \\ 0, & \text{otherwise,} \end{cases}$$

while functional connectivity, using Pearson correlation as an example, is given by

$$A_{\text{functional}}(i, j) = \frac{\text{cov}(x_i, x_j)}{\sigma_{x_i} \sigma_{x_j}}, \tag{1}$$

with $x_i$ and $x_j$ denoting the time series recorded from electrodes $i$ and $j$, respectively. Here, $A_{\text{spatial}}$ captures fixed geometry-driven adjacency based on electrode placement, whereas $A_{\text{functional}}$ encodes data-dependent relationships derived from statistical dependencies in the EEG signals.

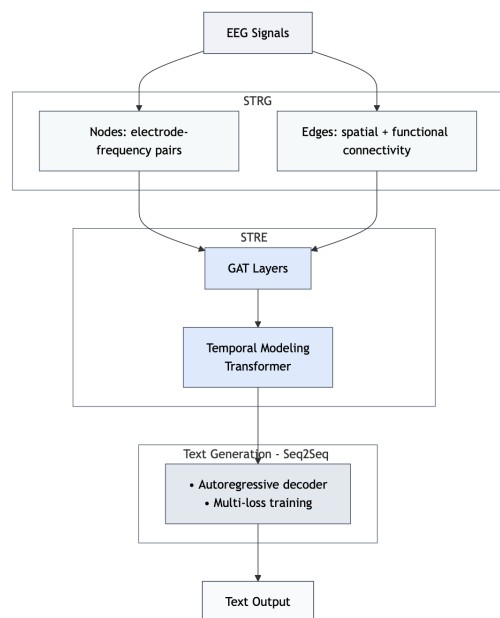

Figure 1: Overview of the proposed Graph-Enhanced framework

### 3.2 SEQUENCE-TO-SEQUENCE DECODING

The objective of EEG-to-text decoding is to map an EEG segment $X$ to a token sequence $Y = (y_1, y_2, \ldots, y_T)$. In the general sequence-to-sequence framework, an encoder processes EEG signals into a latent representation, and a decoder generates text tokens sequentially.

Formally, this task can be formulated as a sequence-to-sequence translation problem:

$$P(Y \mid X) = \arg\max_Y \prod_{t=1}^{T'} P(y_t \mid y_{<t}, X), \tag{2}$$

where $T'$ is the length of the target sentence $Y$, $y_t$ denotes the token at position $t$, and $y_{<t}$ represents the prefix sequence up to position $t - 1$. Here, $X$ denotes the input EEG data, and $P(y_t \mid y_{<t}, X)$ is the conditional probability of generating token $y_t$ given both the preceding tokens and the EEG input. The objective is therefore to maximize $P(Y \mid X)$, the probability of producing the correct target sequence conditioned on the input EEG signals (Liu et al., 2024).

## 4 METHODOLOGY: GRAPH-ENHANCED EEG-TO-TEXT DECODING

### 4.1 OVERVIEW

Our framework consists of three main components: (i) Spectro-Topographic Relational Graphs (STRG) for graph construction, (ii) Spatio-Temporal Relational Embeddings (STRE) for generating graph-aware representations, and (iii) a sequence-to-sequence neural decoder for text generation. As illustrated in Fig. 1, EEG signals are first transformed into STRG to encode spectral and spatial relationships, then embedded into STRE through graph and temporal modeling, and finally decoded into natural language using a Transformer-based decoder.

### 4.2 EEG PREPROCESSING

All EEG recordings are preprocessed using a standardized pipeline implemented in our dataset loader. The preprocessing steps directly follow established EEG cleaning practices and are applied consistently to all samples from ZuCo v1.0 and v2.0.

**High-pass filtering (0.5 Hz).** To eliminate slow drifts and DC offsets, each channel is processed with a fourth-order Butterworth high-pass filter at 0.5 Hz. This step suppresses low-frequency physiological noise and baseline wander, preserving the oscillatory components relevant for downstream analysis.

**Notch filtering (50/60 Hz).** Power-line interference is removed using an IIR notch filter centered at either 50 Hz or 60 Hz, depending on the recording environment. Zero-phase filtering (`filtfilt`) ensures that no phase distortions are introduced, which is essential for temporal alignment.

**Z-score normalization.** To reduce inter-subject and inter-channel variability, each channel is standardized using z-score normalization:

$$x_{\text{norm}}^{(c)}(t) = \frac{x^{(c)}(t) - \mu_c}{\sigma_c + 10^{-8}},$$

where $\mu_c$ and $\sigma_c$ denote the channel-specific mean and standard deviation.

**Frequency band extraction.** Following artifact removal, each EEG window is decomposed into five canonical oscillatory bands using fourth-order Butterworth band-pass filters: delta (0.5–4 Hz), theta (4–8 Hz), alpha (8–12 Hz), beta (13–30 Hz), and gamma (30–100 Hz). Filtering is applied independently per electrode, producing band-specific representations of shape $(C, T)$ that preserve spatial topology and highlight frequency-dependent neural dynamics.

**Sentence-level alignment via wordbounds.** Temporal alignment between EEG and text is achieved using the dataset-provided `wordbounds` files. For each sentence $s_i$, we extract its corresponding EEG interval by converting word-level timestamps into sample indices:

$$[t_{\text{start}}^{(i)}, t_{\text{end}}^{(i)}] = \lfloor \text{wordbound}_i \cdot f_s \rfloor, \quad f_s = 250 \text{ Hz}.$$

If timestamp annotations are partially unavailable, EEG recordings are segmented proportionally across sentences following ZuCo's experimental protocol.

This preprocessing pipeline produces clean, temporally aligned, and frequency-resolved EEG representations that serve as the input to STRG construction.

### 4.3 Spectro-Topographic Relational Graph (STRG)

Given preprocessed EEG signals, we construct STRG to jointly encode spatial topology and dynamic connectivity across electrodes and frequency bands.

**Nodes.** Each node in STRG corresponds to an electrode–frequency band pair, capturing localized spectral features such as band power or wavelet coefficients. This design allows the graph to explicitly represent frequency-specific activity at different scalp locations, providing richer information than treating electrodes or frequency bands in isolation.

**Edges.** Edges are defined from two complementary perspectives. The first is *static spatial adjacency*, which is determined by the physical placement of electrodes. This ensures that electrodes positioned close to each other on the scalp are modeled as graph neighbors. The second is *dynamic functional connectivity*, which is estimated directly from trial-specific EEG signals. Here, measures such as Pearson correlation capture synchronization between electrodes, reflecting dynamic inter-regional interactions that vary with brain state. An illustrative example of the constructed Spectro-Topographic Relational Graph is shown in Fig. 2, where nodes represent electrode–frequency pairs and edges distinguish static spatial adjacency (solid) from dynamic functional connectivity (dashed).

To integrate these two types of relations, we define the weighted adjacency matrix as

$$A = \alpha A_{\text{spatial}} + \beta A_{\text{functional}}, \tag{3}$$

where $A_{\text{spatial}}$ encodes electrode topology, $A_{\text{functional}}$ encodes dynamic connectivity, and $\alpha, \beta$ are hyperparameters controlling their relative influence. This formulation ensures that STRG reflects both the fixed structural organization of the scalp and the context-dependent connectivity patterns in the neural signals.

The graph construction pipeline proceeds as follows. EEG signals are first preprocessed through band-pass filtering and artifact removal, then segmented into word or sentence level windows aligned

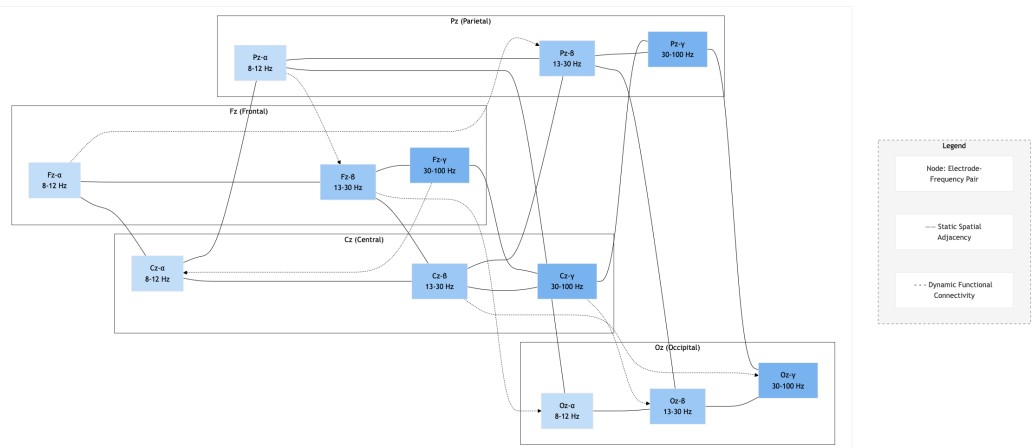

Figure 2: Illustration of Spectro-Topographic Relational Graph (STRG). Each node corresponds to an electrode–frequency pair. Solid edges encode static electrode topology, while dashed edges represent dynamic functional connectivity.

with text. Frequency-domain features are extracted for each electrode, adjacency matrices for spatial and functional relations are computed, and these are combined into the unified adjacency matrix $A$. Finally, the STRG is assembled as $G = \{V, E, A\}$, where $V$ is the set of nodes, $E$ the set of edges, and $A$ the weighted adjacency matrix.

## 4.4 SPATIO-TEMPORAL RELATIONAL EMBEDDINGS (STRE)

STRG captures spectral and spatial relations within each time window, but EEG is inherently dynamic, reflecting transient patterns that evolve across time. To use this temporal nature, we encode STRG with graph neural networks (GNNs) to obtain graph-aware representations, and then model temporal dependencies across windows using sequence models. The resulting spatio-temporal relational embeddings (STRE) serve as the intermediate representation for decoding text. An overview of this stage is shown in Fig. 3.

**Graph encoding.** Given an STRG $G = \{V, E, A\}$ with node features $H^{(0)} \in \mathbb{R}^{N \times d_0}$ (bandpower features extracted from electrode–frequency pairs), we employ Graph Attention Networks (GATs) to encode both spatial and frequency-specific dependencies. At each layer $l$, the embedding of node $i$ is updated as

$$h_i^{(l+1)} = \sigma\left(\sum_{j \in \mathcal{N}(i)} \alpha_{ij}^{(l)} W^{(l)} h_j^{(l)}\right), \tag{4}$$

where $h_j^{(l)}$ is the feature vector of neighbor $j$, $W^{(l)} \in \mathbb{R}^{d_{l+1} \times d_l}$ is a trainable projection matrix, $\alpha_{ij}^{(l)}$ are normalized attention coefficients, and $\sigma(\cdot)$ is a nonlinearity (Ding et al., 2025). The neighborhood $\mathcal{N}(i)$ is defined by the adjacency matrix $A$, which incorporates both electrode topology and dynamic functional connectivity. Unlike a fixed graph convolution, the attention weights $\alpha_{ij}^{(l)}$ adaptively quantify the functional relevance of node $j$ to node $i$. In the EEG context, this means that electrode–frequency pairs exhibiting stronger synchronization or spectral correlation are emphasized, while spurious or noisy relations are downweighted. Stacking multiple GAT layers enables nodes to aggregate information from higher-order neighbors, progressively integrating localized spectral features with broader inter-electrode dependencies. This yields graph-aware representations that preserve frequency-specific activity while capturing the spatial and functional structure of the EEG signals (Demir et al., 2022).

**Temporal modeling.** After graph encoding, each EEG window is summarized by a graph-level embedding $h_t \in \mathbb{R}^d$, obtained through an attention-based readout that aggregates node embeddings within the corresponding STRG. Specifically, the readout operator assigns learnable attention weights to nodes, allowing the model to emphasize electrode–frequency pairs carrying

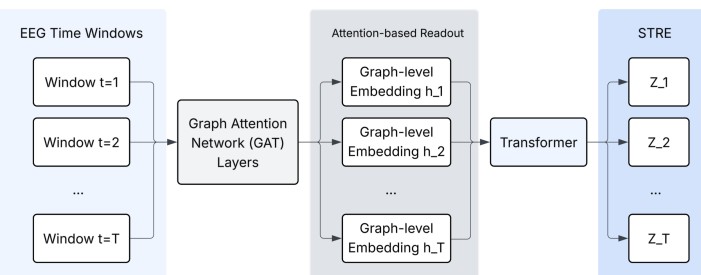

Figure 3: Generation of Spatio-Temporal Relational Embeddings (STRE). GAT layers transform STRG into relational embeddings, which are processed by temporal models (Transformer) to produce contextualized embeddings aligned with text decoding.

stronger spectral or functional relevance. This yields a sequence of window-level representations $\{h_1, h_2, \ldots, h_T\}$ aligned with the temporal order of the EEG signals. To capture dependencies across windows, we employ a Transformer encoder. Unlike recurrent architectures, the Transformer's self-attention mechanism can jointly model both short-range dynamics and long-range dependencies, which are characteristic of EEG. Formally, the temporal encoder produces contextualized embeddings:

$$Z_{1:T} = \text{Transformer}(h_{1:T}), \tag{5}$$

where $Z_t \in \mathbb{R}^{d'}$ denotes the spatio-temporal relational embedding at time step $t$. By stacking temporal self-attention layers, the model integrates information across multiple windows, yielding STRE representations that capture localized spectral features in their temporal context (Li et al., 2023).

**Training Hyperparameters.** The STRG parameters are set to $\alpha = 0.5$ and $\beta = 0.5$. The auxiliary loss weights are fixed to $\lambda_{\text{smooth}} = 0.1$ and $\lambda_{\text{contrastive}} = 0.2$. The STRE encoder uses 2 GAT layers with 4 heads and a 256-dimensional graph embedding. The temporal Transformer contains 4 layers, 8 heads, and a 512-dimensional feed-forward network. The decoder mirrors this architecture. Training uses batch size 16, 50 epochs, AdamW with learning rate $1 \times 10^{-4}$, weight decay $1 \times 10^{-5}$, 1000 warmup steps, and gradient clipping at 1.0.

### 4.5 DECODER

The spatio-temporal relational embeddings (STRE) are mapped into text sequences using an autoregressive Transformer decoder with masked self-attention, which generates tokens step by step conditioned on the encoded STRE and previously predicted tokens. Training this decoder relies on a composite objective. The primary component is the cross-entropy loss, which enforces accurate token prediction by penalizing divergence between the predicted distribution $\hat{y}_t$ and the ground-truth token $y_t$ at each timestep:

$$\mathcal{L}_{\text{CE}} = -\sum_{t=1}^{T} y_t \log \hat{y}_t. \tag{6}$$

To improve robustness and semantic grounding, we introduce two auxiliary losses targeted at EEG-specific challenges. The first is a graph smoothness regularization term that encourages embeddings of adjacent electrode–frequency nodes to remain consistent, thereby suppressing spurious fluctuations across electrodes, expressed as $\mathcal{L}_{\text{smooth}} = \sum_{i,j} A_{ij} \|h_i - h_j\|^2$, where $A$ is the adjacency matrix and $h_i, h_j$ are node embeddings, with a small weighting factor mitigating the risk of over-smoothing. The second is a contrastive alignment loss that bridges the modality gap between EEG and text:

$$\mathcal{L}_{\text{contrastive}} = -\log \frac{\exp(\text{sim}(z, e^+))}{\sum_{e^-} \exp(\text{sim}(z, e^-))}, \tag{7}$$

where $z$ is an STRE representation, $e^+$ its paired text embedding, $\{e^-\}$ a set of negatives sampled from the batch, and $\text{sim}(\cdot, \cdot)$ a similarity function such as cosine similarity. The overall training objective integrates all components compactly as $\mathcal{L} = \mathcal{L}_{\text{CE}} + \lambda_1 \mathcal{L}_{\text{smooth}} + \lambda_2 \mathcal{L}_{\text{contrastive}}$, with $\lambda_1$

and $\lambda_2$ controlling the influence of auxiliary regularization. This joint formulation ensures that the decoder learns not only to predict accurate tokens but also to produce smooth and semantically aligned representations, improving robustness and generalization in noisy EEG settings (Liu et al., 2024).

## 5 EXPERIMENTS

In this section, we evaluate the effectiveness of our proposed graph-enhanced EEG-to-text framework against a set of strong baselines under controlled experimental settings. Our evaluation is designed to test the performance of the model under different training scenarios.

### 5.1 EXPERIMENTAL SETUP

**Datasets.** We use the publicly available ZuCo v1.0 (Hollenstein et al., 2018) and ZuCo v2.0 (Hollenstein et al., 2019) datasets, which provide simultaneous EEG and eye-tracking recordings during naturalistic reading. ZuCo v1.0 contains recordings from 12 subjects reading 1,107 English sentences, with tasks including natural sentence reading and a semantic relation judgment task. ZuCo v2.0 extends this design to 18 subjects with approximately 739 sentences, collected under natural reading and task-specific annotation conditions. Together, these datasets offer a diverse set of experimental paradigms and subject variability, making them well suited for benchmarking EEG-to-text decoding models.

**Training Regularization and Cross-Subject Evaluation.** To mitigate overfitting, we apply dropout (0.1) across all GAT, temporal Transformer, and decoder layers; enable early stopping with patience 5 and min_delta $= 0.001$; use AdamW with weight decay $1 \times 10^{-5}$; and apply gradient clipping (1.0). To evaluate generalization across subjects, we additionally implement a Leave-One-Subject-Out (LOSO) cross-subject evaluation.

**Baselines.** We compare our approach against four representative baselines that capture different modeling paradigms. The biLSTM baseline implements a standard bidirectional LSTM encoder-decoder, representing classical recurrent sequence models (Murad et al., 2025). The BART baseline fine-tunes a Transformer-based seq2seq model, leveraging large-scale pretraining for EEG decoding (El Gedawy et al., 2025). DeWave employs discretized EEG embeddings to capture low-level signal patterns in translation (Duan et al., 2023). Finally, E2T-PTR adopts a contrastive pretraining framework, serving as a recent state-of-the-art baseline (Wang et al., 2024). These baselines collectively span recurrent, Transformer, discretized embedding, and contrastive pretraining approaches, providing a comprehensive reference for comparison.

**Metrics.** We report performance using a combination of lexical overlap and semantic similarity metrics. Lexical evaluation is performed using BLEU (1–4) (Papineni et al., 2002) and ROUGE (Lin, 2004), which measure n-gram precision and recall against reference text. Semantic fidelity is assessed with BERTScore, which leverages contextual embeddings to capture similarity beyond surface-level overlap (Zhang et al., 2020). All models are trained under identical preprocessing pipelines, tokenization procedures, and hyperparameter settings to ensure a fair comparison.

### 5.2 RESULTS

Table 1 compares our proposed Graph-Enhanced EEG-to-Text model against four representative baselines. Across all lexical metrics, our model consistently achieves the highest scores. In particular, it improves BLEU-4 from 8.99 with the strongest baseline (E2T-PTR) to 10.5, marking a relative gain of over 16%. Similar trends are observed for shorter n-gram BLEU scores, where the proposed model outperforms BART and E2T-PTR by 2–3 points on BLEU-1 and BLEU-2. On ROUGE-1-F, our method reaches 34.5, an absolute improvement of nearly two points over the best baseline. Furthermore, the model attains a BERTScore-F of 57.0, substantially higher than BART (53.5), indicating stronger semantic alignment between generated and reference text. These consistent improvements across both lexical overlap and semantic similarity metrics highlight the effectiveness of incorporating graph-based relational modeling into EEG-to-text decoding.

Table 1: Comparison of baseline models with the proposed Graph-Enhanced EEG-to-Text model.

| Model | BLEU-1 | BLEU-2 | BLEU-3 | BLEU-4 | ROUGE-1-F | BERTScore-F |
|---|---|---|---|---|---|---|
| *Baselines* | | | | | | |
| BiLSTM | 38.62 | 21.41 | 11.65 | 6.15 | 27.79 | – |
| BART | 42.34 | 25.26 | 14.91 | 8.89 | 32.66 | 53.53 |
| DeWave | 41.35 | 24.15 | 13.92 | 8.22 | 30.69 | – |
| E2T-PTR | 42.09 | 25.13 | 14.84 | 8.99 | 32.61 | – |
| *Proposed* | | | | | | |
| **Graph-Enhanced** | **45.0** | **27.5** | **16.5** | **10.5** | **34.5** | **57.0** |

Table 2: Ablation study of the proposed Graph-Enhanced EEG-to-Text model.

| Model | BLEU-1 | BLEU-2 | BLEU-3 | BLEU-4 | ROUGE-1-F | BERTScore-F |
|---|---|---|---|---|---|---|
| *w/o Graph* | 38.2 | 21.8 | 10.7 | 7.1 | 27.1 | 52.3 |
| *w/o Static Topology* | 42.7 | 25.0 | 14.8 | 9.3 | 31.8 | 54.1 |
| *w/o Dynamic Connectivity* | 43.1 | 25.6 | 15.2 | 9.7 | 32.4 | 55.0 |
| **Full STRG $\rightarrow$ STRE** | **45.0** | **27.5** | **16.5** | **10.5** | **34.5** | **57.0** |

## 5.3 ABLATION STUDIES

To quantify the contribution of individual components within our framework, we conduct an ablation study, with results reported in Table 2. Removing the graph structure entirely (*w/o Graph*) leads to the largest performance drop, reducing BLEU-4 from 10.5 to 7.1 and BERTScore-F from 57.0 to 52.3. This demonstrates that relational information across electrodes is critical for effective decoding. When retaining the graph but excluding static topology edges (*w/o Static Topology*), performance improves over the sequential baseline but remains lower than the full model, suggesting that anatomical electrode adjacency provides complementary information. Similarly, removing dynamic functional connectivity (*w/o Dynamic Connectivity*) causes a moderate decline, with BLEU-4 falling to 9.7 and BERTScore-F to 55.0. The complete STRG to STRE pipeline achieves the highest scores across all metrics, highlighting the importance of jointly modeling both static and dynamic relationships. Overall, these results confirm that each component contributes meaningfully, and their integration yields the strongest performance.

## 6 CONCLUSION

In this paper, we introduced a graph-enhanced framework for EEG-to-text decoding that explicitly models the spatio-temporal and spectral relationships inherent in brain signals. By constructing Spectro-Topographic Relational Graphs (STRG) and deriving Spatio-Temporal Relational Embeddings (STRE), our approach bridges the gap between sequential EEG recordings and relational brain dynamics. This design moves beyond conventional sequence-only models by integrating static electrode topology with dynamic functional connectivity, thereby offering richer, more interpretable neural representations.

Through comprehensive evaluation on the ZuCo datasets, we demonstrated that our framework consistently outperforms strong recurrent and Transformer-based baselines, achieving up to a sixteen percent relative improvement in BLEU-4. Ablation studies further highlighted the importance of both static topology and dynamic connectivity, confirming that their integration is essential for robust performance. Beyond empirical gains, our results underscore the value of graph-based modeling in enhancing generalization in low-data regimes and in providing neuroscientifically meaningful insights.

Looking forward, our work establishes a foundation for future research at the intersection of graph representation learning, brain–computer interfaces, and natural language processing. Potential directions include scaling the framework to larger and more diverse EEG corpora, integrating multimodal signals such as eye tracking or fNIRS, and exploring interpretable graph-based mechanisms that can shed light on the neural basis of language. By demonstrating the effectiveness of relational graph modeling for EEG-to-text translation, this study opens promising avenues toward more accu-

rate, robust, and interpretable neural decoding systems (Mohammadi & Karwowski, 2025; Graña & Morais-Quilez, 2023; Ju et al., 2024).

## LARGE LANGUAGE MODELS DISCLOSURE

In accordance with the ICLR 2026 policy on responsible use of large language models (LLMs), we disclose that LLMs were used in the preparation of this paper. Their role was limited to aiding in phrasing, polishing, and improving the clarity of writing. The authors take full responsibility for the content of this work.

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
