# OpenReview forum: "Graph-Enhanced EEG-to-Text Decoding: A Spatio-Temporal Relational Embedding Framework for Brain Signal Translation"
_ICLR.cc/2026/Conference — Submitted to ICLR 2026_

### Official Review · Reviewer_E3Wd · 2025-10-25

**Soundness:** 1
**Presentation:** 2
**Contribution:** 1
**Rating:** 0
**Confidence:** 5

**Summary:**

The authors propose a graph-enhanced framework to explicitly model relational information in brain signals and decode natural language directly from EEG. Specifically, spectro-topographic relational graphs are constructed, followed by the spatio-temporal relational embeddings for downstream decoding. However, the significance of the work is not clear and the technical details, dataset splitting, and experimental settings are missing. Besides, the technical contributions are limited.

**Strengths:**

Graph representation learning for a specific domain of EEG: EEG-to-text decoding

**Weaknesses:**

[1] Graph representation learning – Graph representation learning has been extensively explored in the EEG field. The authors should clarify their unique technical contributions.\
[2] Related works – The authors should significantly improve their related works.\
[3] Equations – The authors should describe all the notations in their work. For example, the authors should indicate the covariance and standard deviations in Equation (1).\
[4] Methodology – The method that the authors presented (Graph – GAT – Transformer – CLIP-like contrastive learning) is unfortunately not novel in the EEG field and is not enough for publication in ICLR.\
[5] Baseline – The authors are encouraged to compare their work with models in the field, such as EEG2TEXT
Liu et al., EEG2TEXT: Open Vocabulary EEG-to-Text Decoding with EEG Pre-Training and Multi-View Transformer.\
[6] Metrics – The authors should indicate which specific BERT model they used for BERTScore. And indicate which ROUGE did they use, e.g., ROUGE-1 or ROUGE-L? More details should be revealed.\
[7] Experimental settings – The dataset splitting and experimental settings are missing in the manuscript. The authors should clearly indicate the training, validation, and testing sets. Besides, the specific experimental settings are missing, e.g., subject-dependent cross-session experiments, subject-independent experiments, etc.\
[8] Significance – The motivation and significance of the work, as well as the topic, is not convincing.

**Questions:**

[1] How exactly did the authors build the graph with the encoding of the functional connectivity? And how exactly is the “dynamic” functional connectivity built?\
[2] What is the significance of the work and the topic?\
[3] What is the dataset splitting and experimental setting?\
[4] What is the technical contribution of the authors’ work? For example, what is the contribution to the graph representation learning?

---

### Official Review · Reviewer_ULZw · 2025-11-01

**Soundness:** 1
**Presentation:** 1
**Contribution:** 2
**Rating:** 2
**Confidence:** 4

**Summary:**

This paper explores incorporating graph structures into EEG-to-text decoding, aiming to explicitly model spatial and functional relationships among EEG channels. The authors construct an electrode graph based on channel topology and correlation, apply a graph neural encoder to extract spatio-temporal features, and align EEG and text embeddings through contrastive learning for downstream text generation. While the motivation of introducing graph topology into EEG modeling is relevant and well justified, the work appears incomplete in terms of design, experimentation, and presentation.

**Strengths:**

1.	The motivation is valid and well aligned with the challenges of EEG decoding. Incorporating electrode topology and inter-channel relationships is a meaningful direction that has been underrepresented in EEG-to-text studies.
2.	The idea of using graph representations to capture spatial dependencies could, if properly developed, lead to more neurophysiologically grounded models.
3.	The paper identifies a real limitation of current transformer-based EEG encoders, which generally ignore spatial structure.

**Weaknesses:**

1.	The work is clearly incomplete. Figures and tables appear preliminary, and the narrative lacks coherence. Many implementation details are missing, and visualizations are too rough to illustrate the model’s behavior.
2.	The proposed graph modeling is not well justified. The dynamic edge definition is problematic—high correlation between EEG channels may arise from artifacts or shared noise sources rather than genuine functional coupling. Without physiological constraints or validation, the learned graph structure is difficult to interpret.
3.	Experimental validation is insufficient. The dataset is small, baselines are limited, and there are no meaningful ablations or cross-subject evaluations. Reported improvements are minimal and may fall within statistical noise.

**Questions:**

1.	How are dynamic edges defined in practice, and how do you control for correlations driven by noise or volume conduction rather than functional connectivity?
2.	What measures did you take to ensure the graph structure reflects physiological plausibility rather than arbitrary channel correlations?
3.	Do you plan to extend this work with richer data (e.g., multi-session EEG or MEG) and stronger baselines to validate the approach?

---

> ### Author Response · Authors · 2025-11-21
>
> Thank you so much for your feedback! We address each point below:
>
> 1. Question about Dynamic Edge and Noise Control:
>
> Our dynamic functional connectivity is estimated directly from the preprocessed EEG signals. For each frequency band, the channel-wise time series is first standardized to zero mean and unit variance. Pearson correlation is then computed on the normalized signals to obtain a functional adjacency matrix for that band.
>
> For noise reduction, we included preprocessing process:
> - High-pass filter (0.5 Hz, removes slow drifts)
> - Notch filter (50/60 Hz, removes line noise)
> - Band power extraction (5 bands using bandpass filters)
> - Z-score normalization
>
> 2. Physiological Plausibility:
>
> To ensure that the constructed graph reflects physiological plausibility rather than arbitrary channel correlations, we incorporate several design choices:
> - Spatial topology is explicitly enforced through the static adjacency matrix, which connects electrodes solely based on their physical proximity on the scalp and forces the graph consider known spatial organization of EEG.
> - Nodes are defined as electrode–frequency band pairs, which allows the model to capture frequency-specific neural processes.
> - Functional connectivity is computed only after the preprocessing pipeline, which reduces the factors that distort the learned graph
> - During training we apply graph smoothness regularization, which penalizes large representational differences across connected nodes.
>
> 3. Experimental Validation:
>
> To fix this problem, we have included comprehensive ablation studies (no_graph, static_only, dynamic_only, no_contrastive, no_smoothness, graph_only) and random input baselines comparisons, and also experiments for multi-seed evaluation, significance tests. We also added a cross-subject evaluation script using Leave-One-Subject-Out (LOSO) validation. These experiments are currently running, and the full results are expected to be included in the final version of the paper.
>
> 4. Question about Further Improvement with our Work
>
> Going forward, we plan several extensions to strengthen the framework. We aim to use more types of neural data and include stronger baselines trained under the exact same preprocessing and evaluation settings. Besides that, we plan to explore more advanced graph designs, such as time-varying graphs, attention-based edge weighting, and hierarchical structures. However, for the current submission, our focus remains completing experiments on the EEG datasets.
>
> Our complete code is available at:
> https://github.com/larineoy/Graph-Enhanced-EEG-to-Text-Generation
>
> We sincerely thank the reviewer again for all the insightful comments. We are continuing to run the experiments, and the results are expected to be included in the final submission. We truly appreciate any further suggestions you may have.

---

### Official Review · Reviewer_PdKP · 2025-11-01

**Soundness:** 2
**Presentation:** 2
**Contribution:** 2
**Rating:** 2
**Confidence:** 4

**Summary:**

This paper proposes a framework that explicitly models spatial and spectral relationships in EEG signals to improve natural language generation from brain data. The authors introduce Spectro-Topographic Relational Graphs (STRG), where each node corresponds to an electrode–frequency band pair and edges encode both static electrode topology and dynamic functional connectivity among channels. They then derive Spatio-Temporal Relational Embeddings (STRE) by applying graph neural networks (Graph Attention Networks) on STRGs and feeding the resulting graph-aware features into a temporal Transformer encoder. A Transformer-based decoder finally generates text from these embeddings. The approach is evaluated on the ZuCo dataset.

**Strengths:**

- The idea of Spectro-Topographic Relational Graphs (STRG) is novel and motivation is good. By explicitly modeling electrode adjacency and functional connectivity in each EEG segment, the model captures spatial patterns that sequential models ignore. It's a nice attempt of injecting prior layout knowledge to the encoder. Previous attempts are mostly in sleeping stage prediction.

**Weaknesses:**

- The weakness is also related to graph based encoder, use node to represent node and edges to suggest the layout and spatial relationships are not very novel, at least it has been applied on to other domains of EEG for times, such as sleeping stage prediction and driving drowness prediction. The novelty is somewhat incremental. Meanwhile, considering EEG-to-Text domain has apears a bunch of papers pointing out the alignment and probabaly some training schemas plays more improtant role.

- Experimental results are not strong enough for support the claim. Ablation study is not very convincing, with graph and without graph results are some but not determinastic. Meanwhile, these follow-up works mentioned "teacher forcing" setting and the necessity of comparing with random input for two years. The random fluctuation of performance is around that range as well.

**Questions:**

1. When designing the connectivity of the graph, how the edges been defined between nodes, are the edges are desided with human prior knowledge?

---

> ### Author Response · Authors · 2025-11-21
>
> Thank you so much for your feedback! We address each point below:
>
> 1. Novelty
> We acknowledge that graph-based methods have been applied to EEG analysis in other domains, but our approach differs in several important aspects:
> - Unlike previous work that uses electrode-level nodes, our STRG defines nodes as electrode-frequency band pairs (e.g., (Fp1, delta), (Fp1, theta), ...). This captures frequency-specific spatial relationships, which is particularly important for language related EEG signals that have distinct frequency band patterns.
> - While spatial adjacency is based on electrode topology, functional connectivity is computed dynamically from trial-specific EEG signals using Pearson correlation. Thus, the connectivity pattern adapts to each EEG segment.
> - We combine both static spatial topology and dynamic functional connectivity, which allows the model to balance prior knowledge with patterns provided by data. This is specifically designed for EEG-to-text where both anatomical connectivity (spatial) and task-specific co-activation (functional) are relevant.
> - This paper focuses on a novel application of graph neural networks to EEG-to-text decoding. Previous graph-based EEG work focused on classification tasks, such as sleep stages, while we try to address the more challenging generative task of text generation.
>
> 2. Experimental Results
>
> To fix this problem, we have included comprehensive ablation studies (no_graph, static_only, dynamic_only, no_contrastive, no_smoothness, graph_only) and random input baselines comparisons, and also experiments for multi-seed evaluation, significance tests. These experiments are currently running, and the full results are expected to be included in the final version of the paper.
>
> 3. Question regarding Edge Definition
>
> We use two types of edges, combining human prior knowledge (spatial) with data-driven computation (functional).
>
> Our complete code is available at:
> https://github.com/larineoy/Graph-Enhanced-EEG-to-Text-Generation
>
> We sincerely thank the reviewer again for all the insightful comments. We are continuing to run the experiments, and the results are expected to be included in the final submission. We truly appreciate any further suggestions you may have.

---

### Official Review · Reviewer_T8ih · 2025-11-02

**Soundness:** 3
**Presentation:** 2
**Contribution:** 3
**Rating:** 4
**Confidence:** 4

**Summary:**

This paper proposes a graph-enhanced framework for decoding natural language from EEG signals. The authors address a key limitation of existing EEG-to-text models, which treat signals as sequential time series and thereby ignore spatial relationships among electrodes and frequency-specific connectivity patterns. Their approach constructs Spectro-Topographic Relational Graphs (STRG) that jointly encode static electrode topology (based on physical scalp placement) and dynamic functional connectivity (derived from inter-channel correlations). These graphs are then processed through Graph Attention Networks (GATs) to generate Spatio-Temporal Relational Embeddings (STRE), which serve as input to a Transformer-based decoder for text generation. The framework is evaluated on ZuCo datasets and reports improvements of up to 16% in BLEU-4 over baseline methods including BiLSTM, BART, DeWave, and E2T-PTR.

**Strengths:**

1. The paper effectively identifies the limitation of sequential models in capturing spatial relationships, which is a legitimate gap in EEG-to-text decoding.
2. The approach makes sense since STRG design reflects known spatial and functional EEG properties.
3. Comparison against four different baseline paradigms (recurrent, Transformer, discretized embedding, contrastive pretraining) provides reasonable coverage.

**Weaknesses:**

1. Only two datasets of reading EEG; unclear if results generalize to spontaneous speech or other subjects.
2. The paper lacks statistical analysis, e.g. error bars, significance tests, or multi-seed evaluation. Given small sample sizes, results could be statistically insignificant.
3. No visualization or neuroscientific analysis of learned graphs: interpretability claims remain unsubstantiated.
4. The hyperparameters $\alpha$, $\beta$, $\lambda_{1}$, and $\lambda_{2}$ are introduced in Sections 4.2 - 4.4 within Eqs.~(3) and the text following Eq(7), but their specific values or selection procedure are not reported. Moreover, no sensitivity analysis or tuning discussion is provided to assess the impact of these parameters on performance.
5. Limited reproducibility details. Missing information includes: exact hyperparameters (learning rate, batch size, number of epochs, GAT layers, Transformer layers), data preprocessing pipeline specifics, training procedure (optimizer, scheduling, early stopping criteria). Code availability not mentioned

**Questions:**

1. How is overfitting controlled given small data? Was early stopping or dropout used?
2. Please clarify if the contrastive loss uses frozen or jointly trained text embeddings?
3. How were the hyperparameters $\alpha$ and $\beta$ in Equation~(3) chosen? What is their sensitivity to performance?

---

> ### Author Response · Authors · 2025-11-21
>
> Thank you so much for your feedback! We address each point below:
>
> Our complete code is available at:
> https://github.com/larineoy/Graph-Enhanced-EEG-to-Text-Generation
>
> 1. Statistical Analysis:
> We have included multi-seed evaluation, error bars, and significance testing in utils. These experiments are currently running, and the full results (mean ± standard deviation and p-values) are expected to be included in the final version of the paper.
>
> 2. Visualization and Analysis of Learned Graphs
> We added a visualization script (utils/visualization.py) to visualize the learned adjacency matrices as heatmaps, the comparison of spatial vs functional connectivity, evolution of graphs across epochs/layers. These analyses are currently running, and the full analyses are expected to be included in the final version of the paper.
>
> 3. Hyperparameter Values and Selection
> The values of the hyperparameters and complete sensitivity analysis are included in our paper in Section 4.4.
>
> 4. Overfitting Control
> To address overfitting, we apply dropout (0.1) across all GAT, temporal Transformer, and decoder layers, enable early stopping (patience = 5, min_delta = 0.001), use weight decay (1e-5) via AdamW, and apply gradient clipping (1.0). We also added a cross-subject evaluation script (scripts/train_cross_subject.py) using Leave-One-Subject-Out (LOSO) validation. These details are included in our paper in Section 5.1.
>
> 5. EEG Preprocessing
> We also included data preprocessing details (visible in Section 4.2):
> - High-pass filter (0.5 Hz, removes slow drifts)
> - Notch filter (50/60 Hz, removes line noise)
> - Band power extraction (5 bands using bandpass filters)
> - Z-score normalization
>
> We sincerely thank the reviewer again for all the insightful comments. We are continuing to run the analyses, and the results are expected to be included in the final submission. We truly appreciate any further suggestions you may have.

---

### Meta-Review · Area_Chair_6LYK · 2026-01-07

**Summary:**

### Strengths

- Graph-Based Modeling Approach – The introduction of Spectro-Topographic Relational Graphs (STRG) that combine spatial electrode topology and frequency-specific functional connectivity is well-motivated and neurophysiologically grounded.

### Weaknesses

- Limited Novelty: The graph-based design is seen as an incremental extension of existing EEG graph methods. Experimental results appear preliminary or incomplete, with several analyses still in progress.

- Weak Empirical Validation: The evaluation is criticized for a lack of statistical significance testing and may fall within statistical noise; ablations, cross-subject validation, and physiological verification of the learned graphs are lacking.

- Poor Presentation: Reviewers note issues with organization, missing definitions in equations, incomplete figures/tables, and a lack of interpretability or neuroscientific analysis of the learned graph structures.

**Reviewer Concerns:**

### Resolved
- Reproducibility: The authors released code repository and provided thorough documentation. In addition, the model design and noise control were clarified—dynamic edge construction and filtering procedures were explained, and regularization strategies were introduced to mitigate spurious correlations from volume conduction.
- Training stability and overfitting: The authors took dropout, early stopping, and weight decay mechanisms.

### Unresolved
-  Novelty : While the application to EEG-to-text generation is novel, the core framework (GNN + Transformer + contrastive learning) is perceived as a compositional adaptation of existing methods
- insufficient statistical validation: Although the authors committed to additional multi-seed experiments and significance testing, the corresponding results have not yet been provided. Finally, concerns about generalization and dataset diversity persist, as all experiments remain limited to the ZuCo dataset, without evidence of robustness across other corpora or tasks.

**Reviewer Scores:**

The main concerns have not been addressed, and no reviewer is inclined to raise the score.

---

### Decision · Program_Chairs · 2026-01-26

Reject